# Decision Analytic Modeling for Global Clinical Trial Planning: A Case for HIV-Positive Patients at High Risk for *Mycobacterium tuberculosis* Sepsis in Uganda

**DOI:** 10.3390/ijerph20065041

**Published:** 2023-03-13

**Authors:** Jessica Keim-Malpass, Scott K. Heysell, Tania A. Thomas, Jennifer M. Lobo, Stellah G. Mpagama, Conrad Muzoora, Christopher C. Moore

**Affiliations:** 1School of Nursing, University of Virginia, Charlottesville, VA 22908, USA; 2Division of Infectious Diseases and International Health, School of Medicine, University of Virginia, Charlottesville, VA 22908, USA; 3Department of Public Health Sciences, School of Medicine, University of Virginia, Charlottesville, VA 22908, USA; 4Kibong’oto Infectious Diseases Hospital, Kilimanjaro P.O. Box 447, Tanzania; 5Department of Medicine, Mbarara University of Science and Technology, Mbarara P.O. Box 1410, Uganda

**Keywords:** clinical trial planning decision analysis, decision tree, diagnostic, economic evaluation, HIV, sepsis, tuberculosis

## Abstract

Sepsis is a significant cause of mortality among people living with human immunodeficiency virus (HIV) in sub-Saharan Africa. In the planning period prior to the start of a large multi-country clinical trial studying the efficacy of the immediate empiric addition of anti-tuberculosis therapy to standard-of-care antibiotics for sepsis in people living with HIV, we used decision analysis to assess the costs and potential health outcome impacts of the clinical trial design based on preliminary data and epidemiological parameter estimates. The purpose of this analysis was to highlight this approach as a case example where decision analysis can estimate the cost effectiveness of a proposed clinical trial design. In this case, we estimated the impact of immediate empiric anti-tuberculosis (TB) therapy versus the diagnosis-dependent standard of care using three different TB diagnostics: urine TB-LAM, sputum Xpert-MTB/RIF, and the combination of LAM/Xpert. We constructed decision analytic models comparing the two treatment strategies for each of the three diagnostic approaches. Immediate empiric-therapy demonstrated favorable cost-effectiveness compared with all three diagnosis-dependent standard of care models. In our methodological case exemplar, the proposed randomized clinical trial intervention demonstrated the most favorable outcome within this decision simulation framework. Applying the principles of decision analysis and economic evaluation can have significant impacts on study design and clinical trial planning.

## 1. Introduction

Clinical tuberculosis (TB) caused by the pathogen *Mycobacterium tuberculosis* (Mtb) remains the leading cause of global mortality in patients living with human immunodeficiency virus (HIV). TB deaths among people living with HIV were estimated at 251,000 in 2019 and are a major cause of healthcare utilization in sub-Saharan Africa [1,2,3]. In a pooled analysis of post-mortem studies of hospitalized patients with HIV, the estimated prevalence of TB was 40 percent, which represents a much higher prevalence than that of the general population [3]. Among this sample, TB was disseminated in 88 percent of the cases and was the noted cause of death in 91 percent [3]. 

Microbiological culture for the detection of TB is the gold standard, but it is expensive and has a long wait time [4,5,6]. TB bacteremia is associated with high case fatality rates, but it is not always feasible to culture Mtb from the blood, and sputum samples are often difficult to collect; therefore, many patients presenting with symptoms of sepsis do not receive early anti-TB therapy [7]. The World Health Organization’s recommended standard for persons living with HIV who present with sepsis includes the initiation of anti-TB therapy if a patient is found to have TB based on clinical, microbiological, molecular, or lipoarabinomannan (LAM) antigen tests or empiric initiation if a patient fails to improve after 3–5 days of standard antimicrobial therapy [8]. It is unknown if treatment with immediate anti-TB therapy improves overall survival from sepsis compared with the standard of care (i.e., only treat in the context of positive diagnostics for TB or the failure of standard antibiotics). The determination of efficacy between the two treatment approaches of (1) immediate empiric anti-TB therapy and (2) diagnosis-dependent standard of care is the primary outcome of a proposed randomized clinical trial for people living with HIV and presenting at the hospital with sepsis in Uganda and Tanzania.

Randomized control trials have become increasingly innovative and complex as adaptive designs, multi-center, and multi-country trials become the standard for many diseases including TB and sepsis [9]. Given the resource and human capital investment needed to design, implement, and conduct a global clinical trial, it is critical to understand the potential for short-term efficacy, budget impact of the clinical trial itself, and potential long-term effectiveness following study completion. In global health, the Bill & Melinda Gates Foundation has suggested the use of Highly Efficient Clinical Trials, where investigators take a pragmatic approach to implementation and simulation becomes a natural part of study planning [9]. Given the importance of clinical trial planning in finalizing study interventions, anticipating eligibility criteria, and ensuring scientific success through adequate patient recruitment and enrollment, these components are critical for ethical clinical trial conduct.

Decision analytic modeling is an analytic and simulation approach that integrates and synthesizes sources of evidence in the form of parameter estimates to calculate the cost-effectiveness of applying an intervention. In turn, this estimate can be used to anticipate the impact of the approach on a health system, payer, or society at large, and can be an integral component of clinical trial planning [10]. Additionally, by using this analytic approach to simulate various clinical trial scenarios, it allows for insights into the long-term potential for translation to practice, adoption, and scalability following the trial. 

The purpose of this analysis was to highlight this approach as a case example where decision analysis can estimate the cost effectiveness of a proposed clinical trial design and allow investigators and stakeholders to visualize the potential costs and health impacts based on prior knowledge of expected utilities. In this case, we estimated the impact of immediate empiric anti-TB therapy versus the diagnosis-dependent standard of care using three different TB diagnostics, urine TB-LAM (Alere Determine TB-LAM Ag, Abbot, Chicago, IL, USA) [11,12], sputum Xpert-MTB/RIF (GeneXpert MTB/rifampin (RIF), Cepheid, Los Angeles, CA, USA) [13], and a combination of LAM/Xpert from the perspective of the clinical trial time horizon in Uganda. The time frame of the clinical trial horizon implies limited short-term follow-up and is not representative of the natural dynamics of the entire TB disease course, but can be very helpful in planning resources and potential health impacts prior to clinical trial initiation.

## 2. Methods

### Study Model and Parameters

We constructed three distinct decision-analysis models to assess the cost-effectiveness of immediate empiric anti-TB treatment versus diagnosis-dependent standard of care using the three different TB diagnostic approaches (urine TB-LAM, sputum Xpert MTB/RIF, and LAM/Xpert) (Figure 1 provides one model example using the combined LAM/Xpert diagnostic approach). In our base case for each model, we assumed that the cohort being evaluated met the eligibility criteria of the proposed clinical trial, which included adults ages 18 and older with HIV, admitted to the hospital with a clinical concern for infection, and 2 or more modified quick sepsis-related organ failure assessment (qSOFA) score criteria for sepsis designation [14]. 

Underlying TB prevalence rates in persons with HIV, treatment success and failure rates, and the sensitivity/specificity of urine TB-LAM, Xpert MTB/RIF, and LAM/Xpert are found in Table 1 and were used to guide underlying probability estimates in the model using Bayesian probability estimates. 

Mortality and disability weights (quality adjusted life years, or QALYs) associated with TB/HIV infection, TB treatment, HIV without TB, and sepsis were obtained from the extant literature and applied to the decision tree (Table 1). The cost parameters for the TB diagnostics and TB treatments are shown in Table 1 and were applied from the Ugandan clinical trial perspective to include Ugandan specific costs and prevalence estimates. Importantly, estimates for the efficacy of immediate empiric anti-TB treatments were included based on preliminary data from the investigators. For this analysis from the limited clinical trial perspective and short-term follow-up, we did not account for costs incurred through staff time, consumable supplies, and equipment because those costs can be accounted for in personnel full time equivalents through the clinical trial budget. 

We performed a one-way sensitivity analysis on key parameters by varying each parameter over a range of possible values that were supported by the literature to estimate the effects of parameter uncertainly on the decision analytic cases. In one-way sensitivity analysis, variables were entered into the model with varying estimates based either on known parameter ranges or outcomes greater/less than 10% of the original parameter estimates when a range was not available. We calculated incremental cost effectiveness ratios (ICER) by calculating the differences in costs between early anti-TB treatment and standard care (C1-C0) divided by the differences in effectiveness between early anti-TB treatment and standard care (E1-E0). We conducted the cost effectiveness analysis using TreeAge Pro 2023 R1 software (Williamstown, MA, USA). Institutional review was not required because this is not deemed human subjects research due to the mathematical modeling with parameter estimates obtained from existing published literature.

## 3. Results

Immediate empiric-therapy demonstrated favorable cost-effectiveness compared with all three diagnosis-dependent standard of care models (urine TB-LAM, sputum Xpert MTB/RIF, and the combination LAM/Xpert). Of the three cost effectiveness models, the sputum Xpert diagnostic strategy presented the most favorable incremental effectiveness of early empiric treatment compared with the standard approach (0.08) and ICER (USD 2021.26); followed by urine LAM incremental effectiveness (0.05) and ICER (USD 2683.19); and combined LAM/Xpert incremental effectiveness (0.03) and ICER (USD 4269.73). In sensitivity analysis, the parameter that contributed the most to increased effectiveness of the immediate empiric anti-TB treatment strategy was the proportion of those persons treated with immediate anti-TB treatment who had rifampin susceptible TB, followed by the proportion alive at day seven among those who presented with symptoms of sepsis but who were not found to have diagnostic confirmation of TB, and lastly the proportion of those with rifampin resistance (Figure 2). The proportion alive with untreated TB sepsis did not contribute to the difference in QALYs.

## 4. Discussion

Here, we demonstrated an analytic approach that can provide early proof of concept of the potential costs and health utilities of a proposed randomized control trial demonstrating a mortality benefit to the hypothetical early empiric anti-TB therapy arm regardless of the diagnostic strategy used (urine LAM, sputum Xpert MTB/RIF, or LAM+Xpert). This approach also provided costs estimates and ICER for the time horizon of the clinical trial, which may allow for more precise clinical trial budget estimates [20]. It is already well-established that decision analysis and economic evaluation can guide decision makers on whether strategies designed to improve the quality of life of populations should be scaled up and adopted at a country level [21]. We suggest that using principles of decision analysis and economic evaluation can also have significant impacts on study design when modeled for a very limited clinical trial horizon to help support decisions related to clinical trial design and execution. 

Decision analysis models can help support or refute the underlying hypothesis or treatment modality in question, further refine the budget impact of the trial itself, and differentiate the incremental cost and effect using various diagnostic approaches. Finally, this type of model gave further weight to determining which variables contribute to the overall effectiveness of the intervention, in this case drawing attention to the importance of optimally treating rifampin-susceptible TB. Thus, future strategies should be considered that tailor the composition of the immediate empiric regimen that optimizes rifampin-susceptible TB treatment, such as higher doses of isoniazid and rifampin to overcome potentially suboptimal pharmacokinetics [22,23]. Strategies may also include the optimized testing platforms, such as GeneXpert Ultra and the newer generation of urine LAM assays that have superior sensitivity [24]. There were insufficient clinical performance and cost data to adequately incorporate newer testing modalities such as Fuji SILVAMP TB LAM assay (FujiLAM) into the current model.

### Limitations

We caution that these results and the limited model horizon should not be extrapolated beyond the lens of the clinical trial period, as they only provide early estimates using the preliminary data to support the underlying clinical trial hypothesis, and the demonstrated need of the clinical trial to provide true efficacy and mortality estimates. By incorporating this decision analytic structure prior to enrollment, the same approach can be updated during any pause for interim analysis to update prior parameter estimates. The final decision analytic model can be updated following the completion of the clinical trial using data from the trial itself. Specifically, follow-up decision analysis can then be based on both clinical trial data and real-world epidemiological estimates with a lengthened time horizon (a lifetime horizon with long-term follow up data that includes QALY utility adjustments over a longer period of time) that truly captures the disease burden and trajectory, at the societal- or country-level perspectives. These estimates can also provide the basis for finalized cost estimates, such as those discounted at a 3% annual rate per the guidelines of the Public Health Service Panel on Cost Effectiveness in Health and Medicine or other projections of market pressure and resource allocation constraints [25]. Additionally, instead of data based on the existing literature, we will be able to use probability estimates derived from the clinical trial itself along with more robust Markov models and Monte Carlo simulations based on transitions to various disease states and a greater number of simulations. Additionally, model parameters that can be updated as new empirical data are made available regarding the use of these diagnostics and personalized care in practice [24]. Finally, we can use estimates from our future cost effectiveness analysis to underpin a budget impact analysis for national implementation [26]. 

## 5. Conclusions

In conclusion, decision analysis provides critical insight into the strengths and limitations of approaches in clinical trial design and can provide budget impacts during the clinical trial period for complicated disease states such as TB sepsis in the global clinical trial context. They can also help study teams in the clinical trial planning period visualize the overall potential costs and health impacts when visualizing the model state transitions and potential differences between study arms. In this application, the underlying model parameters derived from the existing literature suggest a novel trial to investigate if the timing of anti-TB therapy for sepsis in people living with HIV may have mortality benefit, and further innovation should focus on the composition of the regimen for the treatment of rifampin-susceptible TB and sepsis without confirmed TB.

## Figures and Tables

**Figure 1 ijerph-20-05041-f001:**
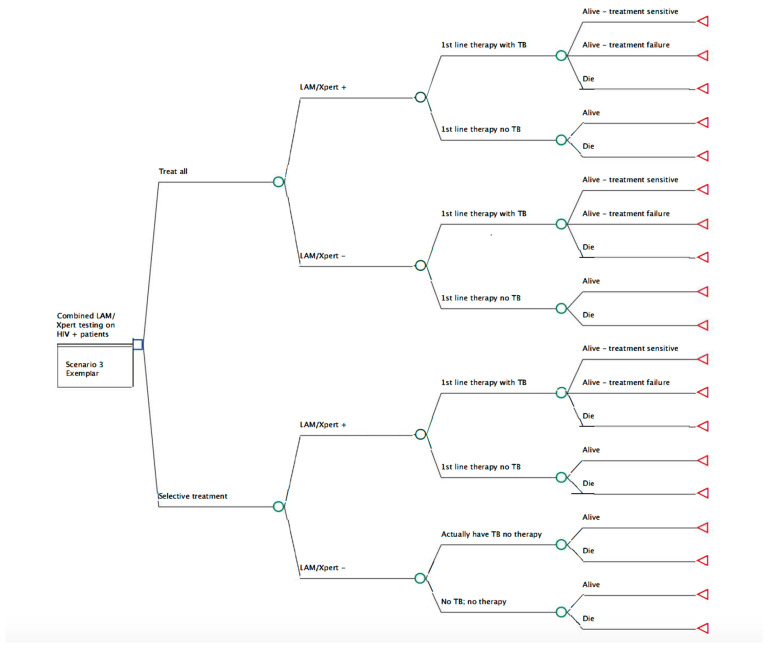
Decision model of combined urine LAM+Xpert. In this model, we compare the two future arms of a randomized control trial. ‘Treat all’ means deliver early antimicrobial therapy in the context of suspected TB-sepsis and ‘selective treatment’ indicates standard of care (wait and treat once a diagnosis is confirmed. This example is a visual representation of the decision tree associated with the combined TB-LAM/Xpert diagnostic testing.

**Figure 2 ijerph-20-05041-f002:**
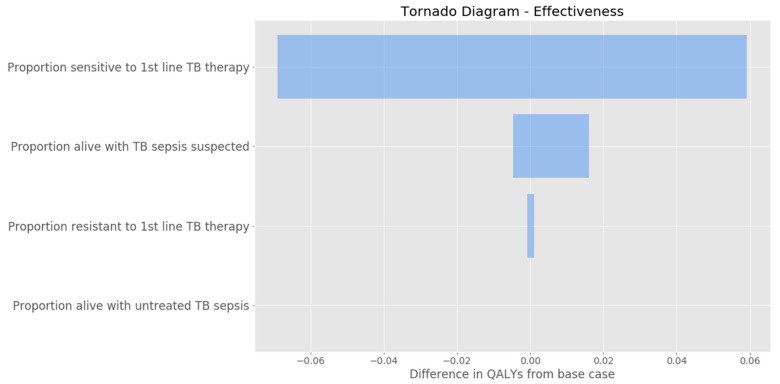
A tornado diagram indicating the sensitivity analysis of input parameters for combined LAM+Xpert. Each line is a variable that is a part of the mathematical model and the blue bar represents their contribution to the difference in QALYs from the base case.

**Table 1 ijerph-20-05041-t001:** Decision analytic model inputs based on data from patients with HIV and sepsis in Uganda.

Diagnostic Abilities	Value (Range if Included in Sensitivity Analysis)	Source
Sensitivity of Urine TB-LAM	53%	Peter et al. [15] Steingart et al. [15]
Specificity of Urine TB-LAM	96%	Peter et al. [15]
Assay sensitivity and ability to obtain diagnostic yield of Sputum Xpert	42%	Gupta-Wright et al./STAMP trial [16,17]
Assay specificity and ability to obtain diagnostic yield of Sputum Xpert	99%	Gupta-Wright et al./STAMP trial [16,17]
Est. sensitivity of combined	63.5%	Lawn et al. [4] Broger et al. [12]
Est. specificity of combined	99%	Shah et al. (Supplement) [6]
Prevalence of TB in HIV + patients with CD4	50%	Shah et al. (Supplement) [6] Broger et al. [12]
Treatment success rate (sensitive)	80% (66–92)	Shah et al. (Supplement) [6]
Treatment failure (resistance)	6.5% (4–15)	Shah et al. (Supplement) [6]
Death in those given TB treatment	13.5%	Unpublished data, Heysell et al. (2020) [18]
Death in those with untreated TB	90% (75–100)	Shah et al. [6]
Death in TB suspect without TB	20% (7–30)	Shah et al. [6]
**Utilities**		
Disability weight TB with HIV infection	0.399	Shah et al. [6]
Disability weight TB treatment	0.1	Shah et al. [6]
Disability weight HIV on ART	0.053	Shah et al. [6]
Disability weight severe sepsis	0.31	Talmor et al. [19]
**Costs**		
Urine LAM	USD 4.19	Shah et al. [6]
Xpert MTB/Rif	USD 17.42	Shah et al. [6]
Treatment costs TB treatment	USD 195	Shah et al. [6]

## Data Availability

Data sharing is not applicable to this article as no datasets were generated or analyzed during the current study. Additional model findings are available upon request.

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
