# Peer review of "Decision Analytic Modeling for Global Clinical Trial Planning: A Case for HIV-Positive Patients at High Risk for Mycobacterium tuberculosis Sepsis in Uganda"

_ijerph, 2023, doi:10.3390/ijerph20065041_

Round 1

Reviewer 1 Report

ijerph-2225740-peer-review-v1

The study presented by Jessica Keim-Malpass and colleagues titled: “Decision analytic modeling for global clinical trial planning: A case for HIV positive patients at high risk for Mycobacterium tuberculosis sepsis in Uganda “investigated how decision analytic modeling, which is a simulating approach that integrates and synthesizes sources of evidence in the form of parameter could help to estimate the cost-effectiveness of an intervention, in this case, pharmacological intervention.  Overall, the paper is well written and the hypothesis and purpose of the study are clearly and concisely presented. In the methods section all the strategy proposed are present and clearly stated. The results are logically presented and in accordance with the significance of the findings. Tables and figures contribute substantially to the content. References are appropriate to the manuscript type, but I suggest adding those references in the discussion 10.1038/s41598-020-59084-2; 10.1093/trstmh/trac079.

Statements in the discussion are clearly supported by data and are linked to goals; however, the author omits the conclusions. Please add it. 

Minor: do not write the title (PhD, MD, RN) in the author section.

Author Response

The study presented by Jessica Keim-Malpass and colleagues titled: “Decision analytic modeling for global clinical trial planning: A case for HIV positive patients at high risk for Mycobacterium tuberculosis sepsis in Uganda “investigated how decision analytic modeling, which is a simulating approach that integrates and synthesizes sources of evidence in the form of parameter could help to estimate the cost-effectiveness of an intervention, in this case, pharmacological intervention.  Overall, the paper is well written and the hypothesis and purpose of the study are clearly and concisely presented. In the methods section all the strategy proposed are present and clearly stated. The results are logically presented and in accordance with the significance of the findings. Tables and figures contribute substantially to the content. References are appropriate to the manuscript type, but I suggest adding those references in the discussion 10.1038/s41598-020-59084-2; 10.1093/trstmh/trac079.

Statements in the discussion are clearly supported by data and are linked to goals; however, the author omits the conclusions. Please add it. 

Minor: do not write the title (PhD, MD, RN) in the author section.

Thank you for your thoughtful review.  We have added the suggested citation in the discussion and we have added a conclusion section.  We have removed the titles in the author section

Reviewer 2 Report

The paper is interesting. However, few comments need to be attended to better understanding to the readers.

COMMENTS TO THE AUTHORS

I would like to thank the authors for taking the time to study about HIV positive patients at a high risk for Mycobacterium. I appreciate the efforts made. However, there are a few comments to consider that might improve your manuscript. 

1. Keywords:

I suggest they be arranged alphabetically.

2. Introduction

Well discussed. However, information on the relationship between HIV and TB is missing. It is important to indicate why this HIV positive patients are said to be at a higher risk of being infected with TB

3. Methodology:

·      In line 92 it is essential to indicate how the sample were selected and data were collected from the three diagnostic dependant standards of care

·      Also indicate what informs the authors in selecting the sample from 18 years and above.

·      Did the authors used patients or patient’s records to access data.

·      Process of getting the permission need to be explained

·      Specific data collection period needs to be indicated

4. Results

·      Not clearly presented

·      Diagram (figure 1) not reader friendly, and it is difficult to follow. I, therefore, suggest that it be restructured for better understanding.

5. Discussion

Discussion needs to be done based on the study findings and supported by relevant literature of which in this case, it is clearly visible except that mostly limitations and recommendations were discussed e.g:

·      Line 148 to 161 seem to be recommendations

·      Line 162 to 164 seem to be limitation

I further recommend that you include the subheading of recommendations and limitation to fit well the above information.

Hoping that the few comments made will assist the authors

Thank you.

Author Response

COMMENTS TO THE AUTHORS

I would like to thank the authors for taking the time to study about HIV positive patients at a high risk for Mycobacterium. I appreciate the efforts made. However, there are a few comments to consider that might improve your manuscript. 

1. Keywords:

I suggest they be arranged alphabetically.

We have updated keywords to be arranged alphabetically.

2. Introduction

Well discussed. However, information on the relationship between HIV and TB is missing. It is important to indicate why this HIV positive patients are said to be at a higher risk of being infected with TB

We included a statement about how HIV positive patients have a disproportionately high percentage of TB cases.

3. Methodology: 

·      In line 92 it is essential to indicate how the sample were selected and data were collected from the three diagnostic dependant standards of care

As we highlight in the manuscript, this was a simulated decision analysis occurring before the randomized controlled trial began with parameter estimates from established literature and epidemiologic studies to simulate.  We also discuss how we will repeat this process after the completion of the randomized control trial to inform a finalized cost effectiveness analysis and translation to care involving a budget impact analysis.  The central premise of the manuscript is that these types of analyses can and should be done prior to the clinical trial as a part of the planning period to anticipate the potential for impact and appropriately budget for the clinical trial.

·      Also indicate what informs the authors in selecting the sample from 18 years and above.

This characteristic was chosen based on the proposed clinical trial eligibility and is stated as such in the manuscript.

·      Did the authors used patients or patient’s records to access data. 

·      Process of getting the permission need to be explained

·      Specific data collection period needs to be indicated

Similar to above, these modelling estimates were based on evidence synthesis from the literature and published epidemiological data.  We are planning/anticipating what the impact of the future clinical trial would be and it’s not until this point (when the clinical trial starts, not a part of this current analysis) that patients will be enrolled and consented. 

4. Results

·      Not clearly presented 

The incremental cost effectiveness ratio (ICER) are the standard reporting for economic evaluation and decision analyses and we have streamlined the results presentation to focus on that.

·      Diagram (figure 1) not reader friendly, and it is difficult to follow. I, therefore, suggest that it be restructured for better understanding.

We present a decision tree here which is often presented in literature involving economic analyses and decision analysis (specifica.  We find that the Figure 1 embedded in the manuscript offers a larger view of it and hope that it makes it easier to follow for readers.

5. Discussion

Discussion needs to be done based on the study findings and supported by relevant literature of which in this case, it is clearly visible except that mostly limitations and recommendations were discussed e.g:

·      Line 148 to 161 seem to be recommendations 

·      Line 162 to 164 seem to be limitation

I further recommend that you include the subheading of recommendations and limitation to fit well the above information.

Have added that suggestion and expanded on the discussion.

Hoping that the few comments made will assist the authors

Thank you.

Thank you for your review of our manuscript!

Reviewer 3 Report

The authors presented case of clinical trial planning on HIV patients with tuberculosis. The estimates of cost-effectiveness of anti-tuberculosis therapy are discussed. The topic raised is important; the work fits to the journal scope. However I have some remarks demanding revision.

Most important is not clear resulting statement. It should be some recommendation which approach is more effective. Currently there is just a common scheme of comparison of cost-effectiveness of treatment and diagnosis-dependent care. The statement “can have significant impact on study design...” (line 28) is not conclusive.

Please update the Abstract -- indicate diagnostics method abbreviations in full - TB-LAM, sputum Xpert-MTB/RIF. Remove the manufacturer names like “(Alere Determine TB-LAM Ag, Abbot, USA)” - it should be in the main text.

Add abbreviation TB for tuberculosis in the Abstract. Or avoid use ‘TB’ in the Abstract.

The keywords list should be extended by specific terms - ‘sepsis’, HIV.

Lines 35 and 40: the citations style: ‘1-3’ , ‘4-6’ - several references are given together without details. Cite one-two references in text together.

The paper is about Uganda case, but in text (line 52) it is East Africa (Uganda and Tanzania), in line 35 - it is sub-Saharan Africa. IT is worthy to rephrase in line 52-53 to be consistent.

Line 78: ‘TB-LAM’ and ‘Xpert-MTB/RIF’ - give these abbreviations in full.

Line 94: qSOFA - mark capital letters to show the abbreviation in wording ‘quick sepsis related organ failure assessment’.

Line 98: the formula is too common. Need add text ‘...where A is ... B is ...’ to denote A and B terms for this particular analysis.

Figure 1 needs more detailed legend.

Remove gray background in Figure 1 graphics. Make larger fonts.

Could add information about number of cases in right part of Figure 1 (where are red triangles)

Table 1 could be updated in the source column: use ‘Shah et al.’ instead of ‘Shah’ citing the papers.

Use word ‘(Supplement)’ in parenthesis for the source ‘Shah Supplement’

“Unpublished data,  Heysell et al. (2020)” - should be in the reference list

“Utilities (QALY)” - here QALY abbreviation us unclear.

Line 102: “Mortality  and disability weights  (QALYs)” - is it different from QALY in the Table?

Line 105: “applied from the Ugandan perspective” - please rephrase. Assume it is based on data from Uganda?  Add details how the parameters were fit.

Line 120: ‘TreeAge Pro’ - add word ‘tool’, add a reference here.

“Institutional review was not required... published data.  ” - it is not correct. Need give reference for these ‘published data’.

Or remove this phrase from the text.

Figure 2 is of low quality. What is the conclusion from the figure? Need indicate if the difference statistically significant or not.

Last line (‘Position alive with unrelated TB sepsis’) is empty.

 Could it be changed to boxplot diagram?

Line 161: ‘GeneXpert Ultra’ - add reference

Line 198: - the phrase (“no dataset”) is not correct. Need at least show dataset as a supplement to the paper

Line 230: “Accessed January 25, 2020.” - old date for publication in 2023.

Please check format of references 13 and 18, and 21 (Uppercase, not full publication data, mixed authors’ initials)

Author Response

The authors presented case of clinical trial planning on HIV patients with tuberculosis. The estimates of cost-effectiveness of anti-tuberculosis therapy are discussed. The topic raised is important; the work fits to the journal scope. However I have some remarks demanding revision.

Most important is not clear resulting statement. It should be some recommendation which approach is more effective. Currently there is just a common scheme of comparison of cost-effectiveness of treatment and diagnosis-dependent care. The statement “can have significant impact on study design...” (line 28) is not conclusive.

We have added additional detail in the abstract supporting the premise and our results.

Please update the Abstract -- indicate diagnostics method abbreviations in full - TB-LAM, sputum Xpert-MTB/RIF. Remove the manufacturer names like “(Alere Determine TB-LAM Ag, Abbot, USA)” - it should be in the main text.

We agree and have removed (they do appear in main text as you suggested).

Add abbreviation TB for tuberculosis in the Abstract. Or avoid use ‘TB’ in the Abstract.

Corrected

The keywords list should be extended by specific terms - ‘sepsis’, HIV.

Added

Lines 35 and 40: the citations style: ‘1-3’ , ‘4-6’ - several references are given together without details. Cite one-two references in text together.

This has been updated as the manuscript transitioned to the typesetting.

The paper is about Uganda case, but in text (line 52) it is East Africa (Uganda and Tanzania), in line 35 - it is sub-Saharan Africa. IT is worthy to rephrase in line 52-53 to be consistent.

Edited to provide consistency.

Line 78: ‘TB-LAM’ and ‘Xpert-MTB/RIF’ - give these abbreviations in full.

Lipoarabinomannan is already spelled out (as is tuberculosis)

Xpert is the trade name of the diagnostic and is not short for anything

Line 94: qSOFA - mark capital letters to show the abbreviation in wording ‘quick sepsis related organ failure assessment’.

Already abbreviated line 114

Line 98: the formula is too common. Need add text ‘...where A is ... B is ...’ to denote A and B terms for this particular analysis.

Removed formula for clarity

Figure 1 needs more detailed legend.

Edited

Remove gray background in Figure 1 graphics. Make larger fonts. 

No gray in Figure 1, larger now that embedded in manuscript.

Could add information about number of cases in right part of Figure 1 (where are red triangles)

This is just a visual representation of one of the 3 scenarios we modeled.

Table 1 could be updated in the source column: use ‘Shah et al.’ instead of ‘Shah’ citing the papers. 

Use word ‘(Supplement)’ in parenthesis for the source ‘Shah Supplement’

“Unpublished data,  Heysell et al. (2020)” - should be in the reference list

Edited and added

“Utilities (QALY)” - here QALY abbreviation us unclear. 

Line 102: “Mortality  and disability weights  (QALYs)” - is it different from QALY in the Table?

Removed the reference in the table and spelled out abbreviation in text

Line 105: “applied from the Ugandan perspective” - please rephrase. Assume it is based on data from Uganda?  Add details how the parameters were fit.

We have added detail to clarify what we mean by this.

Line 120: ‘TreeAge Pro’ - add word ‘tool’, add a reference here.

Added software

“Institutional review was not required... published data.  ” - it is not correct. Need give reference for these ‘published data’.

Or remove this phrase from the text.

Phrase clarified

Figure 2 is of low quality. What is the conclusion from the figure? Need indicate if the difference statistically significant or not.

We have updated the figure to indicate with more clarity what it is representing.  This is a univariate sensitivity analysis (or tornado diagram).  There is no measure of statistical significance.  It is indicating the contribution of individual variables to the overall model in terms of the difference in QALYs from the base case.

Last line (‘Position alive with unrelated TB sepsis’) is empty.

 Could it be changed to boxplot diagram?

As referenced earlier, this parameter is empty because it was not contributing the sensitivity analysis.  A tornado diagram is a specific figure within the cost effectiveness analysis framework and very common in those domains.  A boxplot would not be appropriate here.

Line 161: ‘GeneXpert Ultra’ - add reference

Line 198: - the phrase (“no dataset”) is not correct. Need at least show dataset as a supplement to the paper

No empirical datasets were generated but additional model findings would be available upon request.

Line 230: “Accessed January 25, 2020.” - old date for publication in 2023.

Updated

Please check format of references 13 and 18, and 21 (Uppercase, not full publication data, mixed authors’ initials)

Edited

Round 2

Reviewer 2 Report

Thank you for revising the manuscript and clarifying information which were sounded to be not clearly described. I am happy about the revised version. No further comments

Author Response

Thank you for your thoughtful review.

Reviewer 3 Report

Thank you for the manuscript update. However I have some suggestions to make it more clear for reader with common biological background.

The abbreviations for “urine TB-LAM ...and sputum Xpert-MTB/RIF” are not given in the text (before line 82).

‘LAM’ is only mentioned in line 49, but not RIF.

It is worthy add comments why such methods are important.

For example “Detection of the Mycobacterium tuberculosis cell wall antigen lipoarabinomannan (LAM) in urine permits diagnoses of tuberculosis ...in HIV patients ”

And “Standard treatment regimens for tuberculosis involve prolonged administration of multiple drugs and are usually highly effective. However, Mycobacterium tuberculosis strains that are resistant to one or more of first line drugs require individualized treatment. Rifampin (RIF) resistance is often an indication of multidrug resistance to tuberculosis”.

Please add similar phrases in the beginning to give all the abbreviations used.

Line 101: “probability estimates in the model using Bayes Theorem.” -

Since the formula in line 102 is removed, may not refer to the theorem, but just write

“...Bayesian probability estimates.”

I recommend add a phrase after Figure 1 and before Table 1 to avoid long block of figures without comments (move part of text about Table 1 before the table).

Figure 2 has no any mark for fourth row (Proportion alive with untreated TB sepsis). Either remove this row or put some sign,

The figure title above in graphic (“Tornado Diagram - Effectiveness”) might be removed since it is already in the text.

Finally the reference ‘personal communication” could be numbered.

I believe these remarks could be easily checked not demanding new reviewing round. So, I recommend accept the manuscript now.

Author Response

Thank you for your thoughtful review of our manuscript.

Thank you for the manuscript update. However I have some suggestions to make it more clear for reader with common biological background.

The abbreviations for “urine TB-LAM ...and sputum Xpert-MTB/RIF” are not given in the text (before line 82).

‘LAM’ is only mentioned in line 49, but not RIF.

We added additional abbreviations

It is worthy add comments why such methods are important.

For example “Detection of the Mycobacterium tuberculosis cell wall antigen lipoarabinomannan (LAM) in urine permits diagnoses of tuberculosis ...in HIV patients ”

And “Standard treatment regimens for tuberculosis involve prolonged administration of multiple drugs and are usually highly effective. However, Mycobacterium tuberculosis strains that are resistant to one or more of first line drugs require individualized treatment. Rifampin (RIF) resistance is often an indication of multidrug resistance to tuberculosis”.

We added additional context about the importance of early treatment due to difficulty in obtaining sputum samples reliably.

Please add similar phrases in the beginning to give all the abbreviations used.

Line 101: “probability estimates in the model using Bayes Theorem.” -

Since the formula in line 102 is removed, may not refer to the theorem, but just write

“...Bayesian probability estimates.”

Edited.

I recommend add a phrase after Figure 1 and before Table 1 to avoid long block of figures without comments (move part of text about Table 1 before the table).

Edited

Figure 2 has no any mark for fourth row (Proportion alive with untreated TB sepsis). Either remove this row or put some sign,

The figure title above in graphic (“Tornado Diagram - Effectiveness”) might be removed since it is already in the text.

Added words describing the contribution of proportion alive with untreated TB sepsis.  We think it's important to keep the label at the top so readers can orient themselves to the content displayed.

Finally the reference ‘personal communication” could be numbered.

Edited

I believe these remarks could be easily checked not demanding new reviewing round. So, I recommend accept the manuscript now.